# Moderate-Intensity Constant and High-Intensity Interval Training Confer Differential Metabolic Benefits in Skeletal Muscle, White Adipose Tissue, and Liver of Candidates to Undergo Bariatric Surgery

**DOI:** 10.3390/jcm13113273

**Published:** 2024-05-31

**Authors:** Matías Ruíz-Uribe, Javier Enríquez-Schmidt, Manuel Monrroy-Uarac, Camila Mautner-Molina, Mariana Kalazich-Rosales, Maximiliano Muñoz, Francisca Fuentes-Leal, Carlos Cárcamo-Ibaceta, Daniel J. Fazakerley, Mark Larance, Pamela Ehrenfeld, Sergio Martínez-Huenchullán

**Affiliations:** 1Cardiorespiratory and Metabolic Function Laboratory–Neyün, Valdivia 5090000, Chile; matias.ruiz.uribe@gmail.com; 2Exercise Physiology Laboratory, Faculty of Medicine, Universidad Austral de Chile, Valdivia 5090000, Chile; javier.enriquez@uach.cl (J.E.-S.); manuelmonrroy@uach.cl (M.M.-U.); 3Physical Therapy Unit, Locomotor Apparatus and Rehabilitation Institute, Faculty of Medicine, Universidad Austral de Chile, Valdivia 5090000, Chile; 4Clínica Alemana de Valdivia, Valdivia 5090000, Chile; camilamautner@gmail.com (C.M.-M.); marianap.kalazich@gmail.com (M.K.-R.); maximiliano.cadagan@gmail.com (M.M.); franciscaafuentes@gmail.com (F.F.-L.); ccarcamo@uach.cl (C.C.-I.); 5Surgery Institute, Faculty of Medicine, Universidad Austral de Chile, Valdivia 5090000, Chile; 6Metabolic Research Laboratory, Wellcome-Medical Research Council Institute of Metabolic Science, University of Cambridge, Cambridge CB2 1TN, UK; djf72@cam.ac.uk; 7Charles Perkins Centre and School of Medical Sciences, The University of Sydney, Sydney, NSW 2006, Australia; mark.larance@sydney.edu.au; 8Cellular Pathology Laboratory, Anatomy, Histology, and Pathology Institute, Faculty of Medicine, Universidad Austral de Chile, Valdivia 5090000, Chile; ingrid.ehrenfeld@uach.cl; 9Centro Interdisciplinario de Estudios del Sistema Nervioso (CISNe), Universidad Austral de Chile, Valdivia 5090000, Chile; 10Nephrology Division, School of Medicine, Universidad Austral de Chile, Valdivia 5090000, Chile; 11School of Physical Therapy, Universidad San Sebastián, Valdivia 5090000, Chile

**Keywords:** exercise, bariatric surgery, obesity, insulin resistance, glycemic control, high-intensity interval training

## Abstract

**Background/Objectives:** Bariatric surgery candidates require presurgical physical training, therefore, we compared the metabolic effects of a constant moderate-intensity training program (MICT) vs. a high-intensity interval training (HIIT) in this population. **Methods:** Seventeen participants performed MICT (n = 9, intensity of 50% of heart rate reserve (HRR) and/or 4–5/10 subjective sensation of effort (SSE)) or HIIT (n = 8, 6 cycles of 2.5 min at 80% of the HRR and/or 7–8/10 of SSE, interspersed by 6 cycles of active rest at 20% of the FCR) for 10 sessions for 4 weeks. After training, tissue samples (skeletal muscle, adipose tissue, and liver) were extracted, and protein levels of adiponectin, GLUT4, PGC1α, phospho-AMPK/AMPK, collagen 1 and TGFβ1 were measured. **Results:** Participants who performed MICT showed higher protein levels of PGC-1α in skeletal muscle samples (1.1 ± 0.27 vs. 0.7 ± 0.4-fold change, *p* < 0.05). In the liver samples of the people who performed HIIT, lower protein levels of phospho-AMPK/AMPK (1.0 ± 0.37 vs. 0.52 ± 0.22-fold change), PGC-1α (1.0 ± 0.18 vs. 0.69 ± 0.15-fold change), and collagen 1 (1.0 ± 0.26 vs. 0.59 ± 0.28-fold change) were observed (all *p* < 0.05). In subcutaneous adipose tissue, higher adiponectin levels were found only after HIIT training (1.1 ± 0.48 vs. 1.9 ± 0.69-fold change, *p* < 0.05). **Conclusions:** Our results show that both MICT and HIIT confer metabolic benefits in candidates undergoing bariatric surgery; however, most of these benefits have a program-specific fashion. Future studies should aim to elucidate the mechanisms behind these differences.

## 1. Introduction

Obesity is a global health problem that is increasing its prevalence around the world. Moreover, it is a condition that is known for increasing the risk of developing several cardiometabolic diseases, such as hypertension, insulin resistance, and ultimately type 2 diabetes [1]. Therefore, scientific and public health efforts have been made to counter this phenomenon.

One of the most effective strategies to treat excess body weight is bariatric surgery [2], procedure that has gained popularity among clinicians and the general community because of its high efficiency and low risk of surgical complications and death [3]. However, the physical preparation of the candidate before the surgery has been pointed out as important in order to achieve proper post-surgical weight loss (decrease of fat mass with maintenance or increase of muscle mass) [4,5,6]. More importantly, it has been suggested that improvements in metabolic health can be achieved during the pre-surgical phase in response to the execution of physical training [7]. However, the precise exercise prescription in this population is unknown.

Even when the necessity to prescribe exercise in this population is clear, it is not a simple task for the clinician since only 10% of the candidates for this surgery perform the recommended levels of physical activity [8]. Therefore, it is important to determine the precise benefits of each exercise program and prescribe it according to the candidate’s needs. Aerobic exercise is one of the most frequent types to be prescribed to people with obesity because of its effects on cardiorespiratory fitness and the main utilization of fats as energy sources [9]. However, there are several types of aerobic exercise, where moderate-intensity constant (MICT) and high-intensity interval training (HIIT) are some of the most frequently used [10,11]. Previous pre-clinical work from our group has shown that when MICT and HIIT are compared in the context of obesity, the metabolic benefits of each of them differ depending on the tissue of interest. That is, after 10 weeks of MICT or HIIT in high-fat-fed C57BL/6 mice, heart adiponectin increases [12], and liver collagen reductions were seen only after MICT [13], whereas restoration of adiponectin receptors in skeletal muscle and partial increases of UCP1 in white adipose tissue were seen only after HIIT [13]. These specific effects from MICT and HIIT have been described by other groups as well, where in db/db mice, 10 weeks of HIIT increased GLUT4 levels in the gastrocnemius muscle; however, MICT did not exert any changes [14]. Additionally, even when similar liver insulin-sensitizing effects from MICT and HIIT have been described in diet-induced obese rats, only HIIT reduced NF-κB [15]. 

From a clinical perspective, no known studies have aimed to compare the metabolic effects of MICT and HIIT in the insulin-sensitive tissues of people with obesity, particularly in candidates to undergo bariatric surgery. This comparison is particularly relevant because these two training regimes are frequently used in an interchangeable manner, and the recommendations for these patients in the presurgical phase are broad [16]. Moreover, as previously stated, a cumulative level of evidence suggests that these training programs could confer differential metabolic benefits. Therefore, the aim of this study was to compare the effects on skeletal muscle, white adipose tissue, and liver metabolism of MICT vs. HIIT in candidates for bariatric surgery.

## 2. Materials and Methods

### 2.1. Ethics Statement

This study was reviewed and approved by the Valdivia Health Service Scientific Ethics Committee (Code 350/2020). In addition, all participants read and signed an informed consent before entering this study, and this trial protocol was registered in the International Traditional Medicine Clinical Trial Registry (N° ISRCTN42273422).

### 2.2. Study Design

This study was a randomized controlled trial (Figure 1), where the MICT program was considered the control intervention, given that it is the usual intervention prescribed to these patients. A no-intervention group was not included in this trial as it was considered unethical, given that expert consensus recommends implementing pre-surgical exercise programs in this population [16].

### 2.3. Participants

People that were selected for sleeve gastrectomy, between 18 and 60 years old, were recruited. Exclusion criteria included the inability to attend the exercise sessions, the presence of contraindications to perform physical activity/exercise, functional limitations that could impede the performance of a progressive cardiorespiratory fitness test, and/or the presence of uncontrolled neuropsychiatric illnesses. Randomization was performed by the participants selecting a card with a number, where 1 meant the MICT group and 2 meant the HIIT group. Participants did not know what the number selected meant. The same group of health professionals (i.e., dietitian, psychologist, and surgeon) oversaw all participants to decrease inter-subject variability.

### 2.4. Exercise Intervention and Sample Extraction

For this study, MICT was to exercise for 30 min continuously on a treadmill or cycle ergometer at 50% of the heart rate reserve (HRR) and/or 4–5/10 points in the rating of perceived exertion (RPE) with Borg’s modified scale. Participants in the HIIT group performed 6 bouts of 2.5 min of walking/cycling at 80% of the HRR and/or 7–8/10 RPE, with active rest periods of 2.5 min each at 20% of HRR and/or 1–2 RPE (giving a sum of 30 min) intercalated after each high-intensity bout. A cardiac monitor (Polar (Kempele, Finland)^®^ H10) was used to assess HR during each session, and the physiotherapist managed the speed/inclination or load of the exercise to keep the HR targets. The adherence to the sessions was assessed by the physical therapists in charge of the training session. All participants considered for this study had 100% attendance at the training sessions. After the aerobic exercise was performed, a common strengthening program for both groups was included [16]. This set of exercises focused on the muscles of the abdominal core (horizontal plank), upper (military press), and lower limbs (knee extension, squats) with the following prescription: 3 sets per exercise; intensity at 3 to 5 repetitions in reserve (RIR). The length of both training programs was 4 weeks, where sessions were distributed at a rate of 2 to 3 per week, given that this is the exercise program length recommended by a national expert consensus [16]. After finalizing both training programs, at least 72 h from the last training session to rule out possible lingering acute effects from the last training session, skeletal muscle samples from the transversus abdominis, white adipose tissue from the periumbilical subcutaneous depot, and liver samples were extracted during the bariatric surgery and stored at −80 °C for further analysis.

### 2.5. Phenotypical Measurements

At the beginning and after the completion of the training programs, height, body weight, BMI, waist and hip circumferences, waist-to-hip and waist-to-height ratios were measured (FF), and through bioimpedance (InBody (Seoul, Republic of Korea)^®^ 270) [17], the body composition was assessed. The International Physical Activity Questionnaire, as recommended by national guidelines [16], was used to indirectly measure the physical activity levels (excluding the exercise sessions performed for this study) of the last 7 days. Grip strength was assessed with a Jamar^®^ digital hand-held dynamometer in a seated position, where the participant had to have their shoulder in neutral position, elbow in 90° flexion, and wrist in neutral position. The best of three tries was considered for further analysis. 

### 2.6. Laboratory Exams

At the beginning and end of the training programs, and at least 72 h after the last training session, venous blood samples were collected from the antecubital vein in the morning and after at least 8 h of fasting. Afterwards, the samples were transferred to the Clinical Laboratory of the Universidad Austral of Chile, where lipid profiles (Colestat and TG color, Wiener Lab (Rosario, Argentina)^®^), transaminases, glycemia (based on glucose oxidase/peroxidase activity), insulinemia (specific fluorometric enzymatic assay ST AIA-PACK IRI, Tosoh (Tokyo, Japan)^®^), HbA1c (cation-exchange high-performance liquid chromatography in an automated analyzer), TSH (specific immunoassay), and kidney function markers (creatinine, uric acid, urea, and blood uric acid) were measured. As an approximation of insulin resistance from a clinical perspective, HOMA-IR were calculated using fasting and after 120 min of glucose ingestion values of glycemia and insulinemia, using the following formula: (insulinemia (µUI/mL) * glycemia (mmol/L))/405.

### 2.7. Protein Analysis by Western Immunoblotting and Histology

Transversus abdominis muscle (25–30 mg), liver (20–25 mg), and white adipose tissue (100–150 mg) were homogenized with an electronic hand-held homogenizer (D1000 homogenizer, Benchmark Scientific (Sayreville, NJ, USA)^®^) in ice-cold radioimmunoprecipitation assay buffer (RIPA, Abcam (Cambridge, UK)^®^ catalog number ab156034). Proteins (30 µg for muscle and liver samples and 20 µg for white adipose tissue) after extraction and quantification using a bicinchoninic acid (BCA) protein assay (Visual Protein (Taipei City, Taiwan)^®^ catalog number BC03-500) were run on polyacrylamide gradient gels (4–15%) (Bio-Rad (Hercules, CA, USA)^®^, catalog number 4568086). Then, the transference of these proteins from the gels to a nitrocellulose membrane (Bio-Rad^®^, catalog number 1704158) was performed using the Trans-Blot Turbo™ Transfer System (Bio-Rad^®^, Hercules, CA, USA). This was followed by 1 h of incubation using 5% bovine serum albumin (BSA) or 5% skim milk in buffer (TBST) containing 0.6% Tris HCl *w*/*v*, 0.1% Tris-base *w*/*v*, 0.6% NaCl *w*/*v*, and 0.05% Tween-20 *v*/*v*. Membranes were then washed in TBST buffer (3 × 10 min, and incubated with primary antibodies overnight at 4 °C as follows: Peroxisome proliferator-activated receptor-gamma coactivator (PGC)-1α (1:1000, rabbit monoclonal Cell Signaling (Danvers, MA, USA)^®^, catalog number 2178S), GLUT4 (1:1000, mouse monoclonal Cell Signaling^®^, catalog number 2213S), adiponectin (1:1000, rabbit monoclonal Cell Signaling^®^, catalog number 2789S), adiponectin receptor (ADIPOR) 1 (1:2000, rabbit monoclonal Novus Biologicals (Centennial, CO, USA)^®^, catalog number NBP2-67631), collagen 1 (1:1000, rabbit monoclonal Cell Signaling^®^, catalog number 39952S), transforming growth factor beta (TGFβ) 1 (1:2000, rabbit monoclonal Novus Biologicals^®^, catalog number NBP1-80289), phospho-AMPK (1:1000, rabbit monoclonal Cell Signaling^®^, catalog number 2535S), AMPKα (1:1000, rabbit monoclonal Cell Signaling^®^, catalog number 2532S), and uncoupling protein (UCP) 1 (1:1000, rabbit monoclonal Novus Biologicals^®^, catalog number NBP2-20796). At the end of this incubation, membranes were washed in TBST buffer (3 × 10 min) and incubated for 1 h at room temperature in a buffer with a secondary antibody labeled with peroxidase (1:2000, anti-rabbit IgG, Cell Signaling^®^, catalog number 7074S; or 1:2000, anti-mouse IgG, Cell Signaling^®^, catalog number 7076S). After incubation with the secondary antibody, membranes were washed with TBST buffer and incubated briefly with a chemiluminescent substrate (ClarityTM Western ECL substrate, Bio-Rad^®^, catalog number 170-5061) for visualization on a G:Box system (Syngene (Bangalore, India)^®^). Densitometric analysis of the bands was performed with the ImageJ software version 1.53t. Protein loading normalization was performed using Ponceau S staining of the whole membrane in each respective experiment.

Part of each sample was separated and fixated in 10% formalin for a minimum of 24 h and underwent overnight processing at the Nephrology Laboratory of the Hospital Base Valdivia. From paraffin-embedded blocks, 5 μm sections were stained with picro-sirius red (Abcam^®^, catalog number ab150681) to investigate the impact of MICT and HIIT on muscle, liver, and white adipose tissue structure.

### 2.8. Statistical Analysis

The mean and standard deviations were calculated to summarize the quantitative data, whereas absolute frequencies were used to exhibit qualitative data. Before inferential statistical analysis, the data distribution was assessed through the Kolmogorov–Smirnov test. For inter-group comparisons, the Mann–Whitney test was used, whereas the intra-group comparisons were performed with Wilcoxon’s test. For all analyses, a *p* < 0.05 was considered statistically significant, and the software used for these analyses was SPSS version 20 (IBM (Armonk, NY, USA)^®^) and GraphPad v8.

## 3. Results

Out of 38 candidates that were screened, 17 (MICT = 9; HIIT = 8) finished the exercise sessions and accepted the sample extraction procedure (Figure 1). In terms of the phenotypical effects of both training programs, MICT promoted slight but significant changes in body weight and composition, whereas HIIT decreased waist and hip circumferences as markers of central obesity (Table 1). No major differences in terms of body weight loss were seen between groups (*p* > 0.05).

After MICT, markers of kidney stress increased (e.g., creatinine, urea, and blood uric nitrogen), whereas HIIT promoted a small and significant decrease in HbA1c (Table 2).

In the skeletal muscle samples, higher levels of PGC-1α were observed only after MICT (Figure 2), where no major changes were seen in GLUT4, adiponectin, or ADIPOR1. In the liver, histological images showed higher signs of collagen deposition in the participants from the MICT group, a finding that was associated with a higher quantity of collagen 1 from these patients, with no changes in TGFβ1 (Figure 3). 

Interestingly, MICT participants had higher liver levels of phospho-AMPK/AMPK and PGC-1α (Figure 4). Lastly, in the white adipose tissue, higher levels of adiponectin were seen in the HIIT group, with no differences between groups regarding ADIPOR1 and UCP1 (Figure 5).

## 4. Discussion

This study aimed to compare the effects on skeletal muscle, white adipose tissue, and liver metabolism of a constant moderate-intensity training program (MICT) vs. a high-intensity interval training (HIIT) in candidates for bariatric surgery. Our main findings were that both training programs confer metabolic benefits, even with mild changes in body composition; however, those benefits differ depending on the tissue and proteins of interest, highlighting the specificity of the metabolic effects of exercise in the context of obesity.

The metabolic effects of exercise on skeletal muscle metabolism have been a fruitful scientific field in the past decades, where PGC-1α has been described as a key regulator of insulin sensitivity [18] and mitochondrial biogenesis [19]. In our study, we found that MICT was more effective at increasing this protein compared to HIIT, a finding that is somewhat conflicting to what other groups have described. For instance, Egan et al., after comparing the acute effects of a single session of low- or high-intensity exercise training in sedentary men, found that the mRNA levels of PGC-1α of the vastus lateralis were higher [20], which was also correlated with higher levels of phospho-AMPK/AMPK ratios. In line with previous in preclinical studies, 8 weeks of HIIT has shown positive effects on increasing skeletal muscle PGC-1α in a mouse model of type 2 diabetes [21]. However, our finding seems not to be random, given that the participants that underwent MICT also significantly improved their body composition, which has been previously related to the upregulation of PGC-1α in people with obesity [22]. We hypothesize that the divergent results that we saw might indicate that the degree of obesity could impact how skeletal muscle adapts to exercise training. This idea is supported by the findings of Ryan et al., given that when they compared the effects of 12 weeks of MICT and HIIT on the insulin sensitivity of adults with obesity, they described similar effects after both training programs [23], highlighting the possibility that when obesity is involved, the impact of the exercise intensity is not the same as when analyzing healthy subjects [24]. The absence of changes in terms of GLUT4 is somewhat expected, given that most of the exercise effects on this protein are focused on the location in the cell, where translocation of these receptors from the cytoplasm to the nucleus has been widely described [25,26], and this effect is transient for hours after the last training session [27], and the samples from this study were collected no less than 72 h after training cessation. From the best of our knowledge, this is the first study that reports the presence of adiponectin in skeletal muscle in a clinical setting, even when preclinical studies from our group [13,28] and others [29,30,31] have described and suggested its exploration in human samples. Independently of the absence of differences between training programs, the question regarding the effects of exercise itself on this protein remains since no untrained control group was included in this trial, along with the levels of its receptor ADIPOR1, considering that in preclinical studies we have seen obesity-related downregulations of both proteins and partial restoration after exercise [13,28].

The effects of obesity on the liver metabolism and structure have been a prolific field for research given its susceptibility to store ectopic fat and the development of metabolic dysfunction [32], along with the potential reversal (at least partially) effects of exercise [33]. In our study, we found that participants who performed HIIT exhibited lower levels of collagen deposition, findings that are conflicting compared to recent literature. For instance, Houttu et al., in a recent systematic review and meta-analysis, described that both MICT and HIIT had similar benefits for reducing intrahepatic lipids in humans [34]. On the other hand, 12 weeks of HIIT decreased liver stiffness (measured through FibroScan) in sedentary men with nonalcoholic fatty liver disease, whereas MICT did not confer major benefits in this outcome [35], findings that are in agreement with others [36]. In terms of the mechanistic reasons behind these differences between programs, we explored the levels of TGFβ, a known growth factor that regulates the liver extracellular matrix [37]. Nevertheless, no major changes were seen after both training programs. This could indicate that other proteins that regulate the extracellular matrix might be involved in this differential response, such as connective tissue growth factor and the balance between matrix metalloproteinases and their inhibitors (TIMPs) [38]. From a metabolic perspective, interestingly, after MICT, higher levels of phosphor-AMPK and PGC-1α were seen, which are somewhat counterintuitive to what we saw in terms of structure. One possible explanation is that HIIT training is decreasing the metabolic burden on the liver, by which this organ does not require higher levels of mitochondrial function to compensate for the excess of lipids in the context of fatty liver disease. This hypothesis is supported by the findings of Koliaki et al. because, when they compared patients with fatty liver vs. lean volunteers, the former had 4.3 to 5.0-fold higher maximal respiration rates in their isolated mitochondria [39]. However, this hypothesis requires further investigation given that no specific experiments on the mitochondria of the liver samples were performed.

Adipose tissue fibrosis in the context of obesity is one of the proposed mechanisms behind the metabolic dysfunction of this tissue in the context of obesity and insulin resistance [40,41,42]. For this reason, we explored the structure of this tissue after MICT and HIIT, where no major differences between groups were seen. In terms of adiponectin production, we saw a differential response between programs, where participants after HIIT exhibited higher levels of adipose tissue adiponectin. Most of the previous studies in the field have described positive effects from exercise by increasing circulatory adiponectin in patients with metabolic dysfunctions (i.e., type 2 diabetes) [43], but to the best of our knowledge, this is the first study that explores the effects of two different exercise programs on this protein directly in the tissue; therefore, this differential response between programs could be one of the reasons behind the variability of this response in people with obesity [44]. Mechanistically, we hypothesize that HIIT confers this specific adiponectin increases because of the higher insulin production during high-intensity exercise compared to moderate-intensity training [45]. Given that previous studies have pointed out that insulin stimulates adiponectin production in adipocytes [46], nevertheless, future studies should explore the mechanisms behind this differential response. No major changes were seen in terms of adiponectin receptor 1 and UCP1 as a browning marker, a finding that is concurrent with others, particularly in clinical experiments where exercise fails to increase browning markers in adipose tissue in the context of obesity [47] or activate brown adipose tissue [48].

This study has several limitations. For instance, no untrained group was included in this study, particularly for ethical reasons given that all candidates should undergo a physical preparation prior to surgery; however, this hinders the possibility of assessing the effects of both training programs in the absence of exercise. Secondly, most of the volunteers were women, which could bias the results towards one gender. Thirdly, for practical reasons, we were not able to collect tissue samples prior to the exercise training to have a clearer image of the baseline status of the adipose tissue, liver, and skeletal muscle. Nevertheless, we tried to counter those limitations by randomly conforming the groups and therefore securing baseline similarity between them.

## 5. Conclusions

Both MICT and HIIT confer metabolic benefits in candidates to undergo bariatric surgery; however, most of these benefits have a program-specific fashion, where MICT seems to be more beneficial to skeletal muscle metabolism, whereas HIIT seems to improve several metabolic markers in the liver and white adipose tissue. However, because of the nature of this study, it is not possible to identify the possible mechanisms behind these adaptations, which should be explored in future studies.

## Figures and Tables

**Figure 1 jcm-13-03273-f001:**
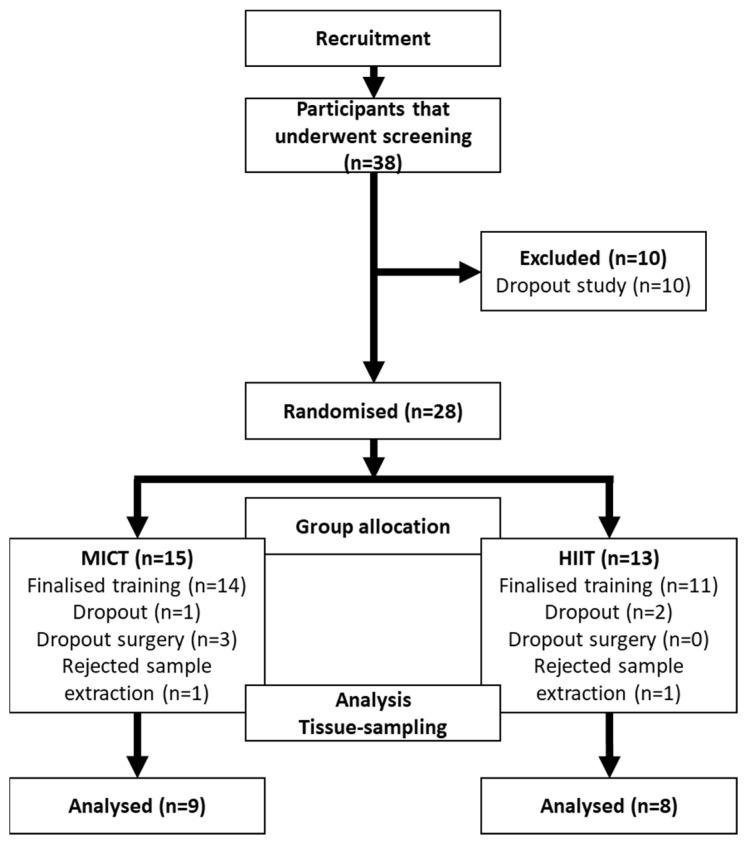
CONSORT model.

**Figure 2 jcm-13-03273-f002:**
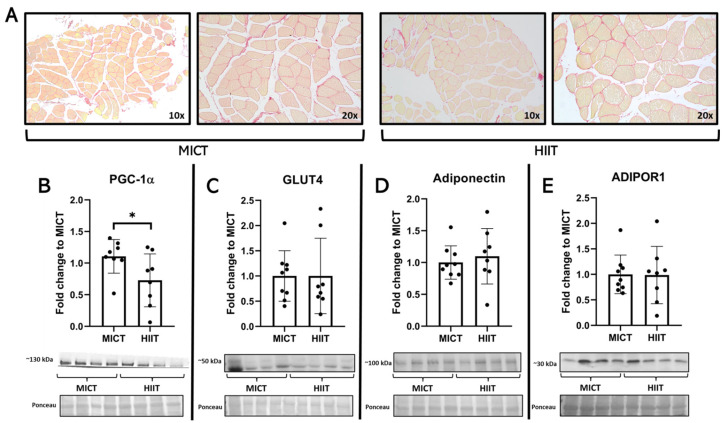
Effects of MICT and HIIT on skeletal muscle. (**A**) Representative histological images stained with Picro Sirius Red at 10× and 20×. (**B**) Peroxisome proliferator-activated receptor-gamma coactivator (PGC)-1α; (**C**) GLUT4; (**D**) muscle adiponectin; and (**E**) adiponectin receptor (ADIPOR) 1. Their representative blots are below its respective graph with its membrane stained with Ponceau S used as loading control. Data are presented as mean ± SD. *: *p* < 0.05. Each dot represents a participant.

**Figure 3 jcm-13-03273-f003:**
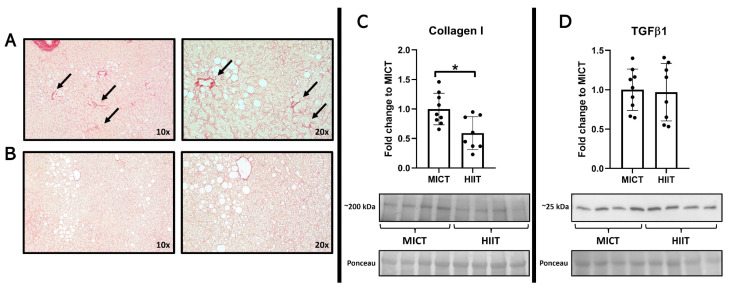
Effects of MICT and HIIT on the liver. (**A**) Representative histological images stained with Picro Sirius Red at 10× and 20× for (**A**) MICT group and (**B**) HIIT group. Arrows indicate collagen accumulation points. (**C**) Collagen 1 and (**D**) transforming growth factor (TGF) β1 and their representative blots below its respective graph with its membrane stained with Ponceau S used as loading control. Data are presented as mean ± SD. *: *p* < 0.05. Each dot represents a participant.

**Figure 4 jcm-13-03273-f004:**
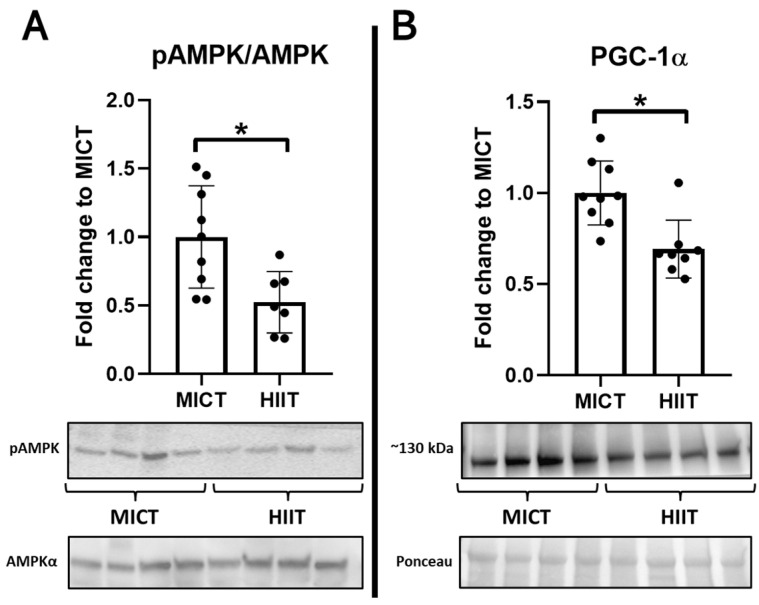
Effects of MICT and HIIT on the liver (2). (**A**) Phospho-AMPK/AMPK ratios, (**B**) peroxisome proliferator-activated receptor-gamma coactivator (PGC)-1α, and their representative blots below its respective graph with its membrane stained with Ponceau S used as loading control. Data are presented as mean ± SD. *: *p* < 0.05. Each dot represents a participant.

**Figure 5 jcm-13-03273-f005:**
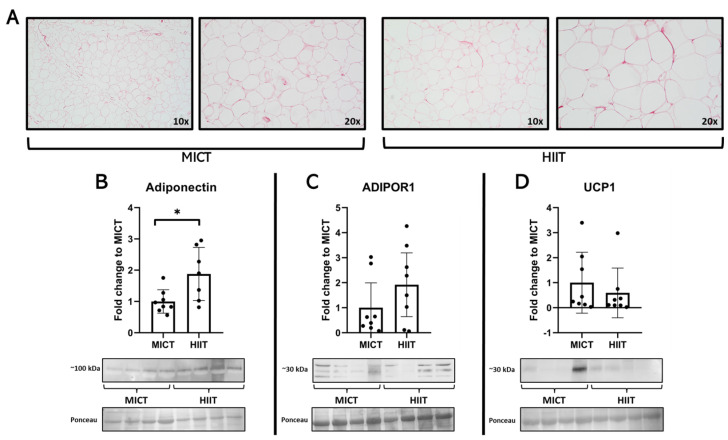
Effects of MICT and HIIT on the white adipose tissue. (**A**) Representative histological images stained with Picro Sirius Red at 10× and 20×. (**B**) Adiponectin, (**C**) adiponectin receptor (ADIPOR) 1, (**D**) uncoupling protein (UCP) 1 and their representative blots below its respective graph with its membrane stained with Ponceau S used as loading control. Data are presented as mean ± SD. *: *p* < 0.05. Each dot represents a participant.

**Table 1 jcm-13-03273-t001:** General characteristics of the participants.

Parameter	MICT (n = 9)		HIIT (n = 8)	
	Pre	Post	Pre	Post
Sex (F/M)	8/1		7/1	
Age (years)	36.1 ± 6.7		33.3 ± 8.3	
Height (m)	1.62 ± 0.08		1.65 ± 0.08	
Weight (kg)	104 ± 14	101 ± 14 *	109 ± 18	107 ± 19
BMI (kg/m^2^)	41 ± 4.8	38 ± 2.9 *	40 ± 6.6	39 ± 6.8
Fat mass (%)	48 ± 3.1	46 ± 3.2 *	46 ± 6.5	48 ± 4.6
Muscle mass (%)	29 ± 1.8	30 ± 1.9 *	29 ± 3.1	29 ± 2.7
Waist circumference (cm)	114 ± 13	110 ± 14 *	124 ± 16	117 ± 17 *
Hip circumference (cm)	124 ± 7	121 ± 4	129 ± 8	126 ± 9 *
Waist-to-hip ratio	0.92 ± 0.09	0.90 ± 0.10	0.96 ± 0.08	0.93 ± 0.08 *
Waist-to-height ratio	0.71 ± 0.08	0.68 ± 0.08 *	0.75 ± 0.11	0.71 ± 0.11 *
Spontaneous physical activity (MET × min × week)	1189 ± 754	1984 ± 883	934 ± 791	2039 ± 1221
Sitting hours per day (n)	5.5 ± 2.8	4.9 ± 2.4	5.8 ± 3.3	4.8 ± 1.7
Grip strength (kg)	33.7 ± 7.1	34.1 ± 7.3	30.5 ± 6.4	30.6 ± 5.4
Comorbidities (n) Hypertension Hypothyroidism Insulin resistance Type 2 diabetes Anxiety Depression None (only obesity)				
	2		3	
	2		2	
	3		2	
	0		0	
	1		1	
	0		1	
	4		4	

Abbreviations: BMI = body mass index; MICT = moderate-intensity constant training; HIIT = high-intensity interval training. Values expressed in frequencies or mean ± SD. *: statistical difference intra-group by Wilcoxon’s test.

**Table 2 jcm-13-03273-t002:** Laboratory outcomes by group.

	MICT (n = 9)		HIIT (n = 8)	
Parameter	Pre	Post	Pre	Post
Cholesterol (mg/dL)	177 ± 22	170 ± 22	183 ± 51	185 ± 55
Triglycerides (mg/dL)	126 ± 52	144 ± 68	148 ± 123	118 ± 63
HDL (mg/dL)	40 ± 12	38 ± 12	47 ± 10	47 ± 10
LDL (mg/dL)	112 ± 24	103 ± 23	112 ± 40	114 ± 41
VLDL (mg/dL)	25 ± 10	29 ± 14	30 ± 24	23 ± 13
No-HDL (mg/dL)	138 ± 23	132 ± 14	136 ± 49	138 ± 51
Albumin (g/dL)	4.3 ± 0.19	4.3 ± 0.17	4.3 ± 0.20	4.4 ± 0.22
Uric acid (mg/dL)	5.2 ± 1.5	5.2 ± 1.5	4.6 ± 1.4	4.5 ± 1.0
Creatinine (mg/dL)	0.74 ± 0.11	0.82 ± 0.12 *	0.77 ± 0.11	0.77 ± 0.14
Urea (mg/dL)	27 ± 7	36 ± 12 *	27 ± 8	26 ± 7
Blood uric nitrogen (mg/dL)	12 ± 3.3	17 ± 5.6 *	13 ± 3.8	12 ± 3.0
GOT (UI/L)	22 ± 6.3	31 ± 36	21 ± 6.8	23 ± 10
GPT (UI/L)	29 ± 11	46 ± 51	29 ± 14	33 ± 20
Fasting blood glucose (mg/dL)	98 ± 11	98 ± 10	93 ± 14	94 ± 11
Fasting insulin (µUI/mL)	19 ± 8	21 ± 16	20 ± 10	18 ± 7
HOMA-IR	4.6 ± 1.8	5.3 ± 4.5	4.7 ± 2.4	4.1 ± 1.6
HbA1c (%)	5.6 ± 0.6	5.5 ± 0.5	5.5 ± 0.4	5.4 ± 0.4 *
TSH (µUI/mL)	2.3 ± 1.7	2.7 ± 1.5	2.4 ± 1.6	2.2 ± 1.2

Abbreviation list: BMI = body mass index; MICT = moderate-intensity constant training; HIIT = high-intensity interval training; HDL = high-density lipoprotein; LDL = low-density lipoprotein; VLDL = very low-density lipoprotein; GOT = glutamic oxaloacetic transaminase; GPT = glutamic pyruvic transaminase; HOMA-IR = homeostatic model assessment for insulin resistance; HbA1c = hemoglobin A1c; TSH = thyroid stimulating hormone. Values expressed in frequencies or mean ± SD. *: statistical difference intra-group by Wilcoxon’s test.

## Data Availability

Data are available upon request to the corresponding author at sergio.martinez@uss.cl.

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
