# Peer review of "Moderate-Intensity Constant and High-Intensity Interval Training Confer Differential Metabolic Benefits in Skeletal Muscle, White Adipose Tissue, and Liver of Candidates to Undergo Bariatric Surgery"

_jcm, 2024, doi:10.3390/jcm13113273_

Round 1

Reviewer 1 Report

Comments and Suggestions for Authors

The authors studied the metabolic effect of 2 types of presurgical physical training like MICT and HIIT in bariatric surgery candidates, pathologically examining the tissue samples for skeletal muscle, white adipose tissue and liver. There are several points to be addressed by the authors.

First, the authors could tell why they chose 4-week duration of training before surgery that should have some rational reasoning; i.e. sufficiently long to show anticipated metabolic effect. Moreover, why the tissue samples were extracted 72 hours after the last training session? The time interval could affect the findings.

Second, the authors could tell how they assessed the adherence of the participants to both types of training program.

Third, in page 3 the authors described that “skeletal muscle samples from the transversus abdominis ------ were extracted” but in page 4 “Quadriceps muscle ------ were homogenized”. Which one is correct?

Fourth, the authors could tell or comment on the change in serum creatinine, urea and blood uric nitrogen only in the MICT group, and please explain about what is blood uric nitrogen.

Fifth, the authors could tell whether there might be any difference in body weight loss or metabolic outcomes after surgery between 2 groups of participants with obesity which underwent MICT and HIIT.

Author Response

Dear reviewer,

Reviewer 2 Report

Comments and Suggestions for Authors

This study was to compare the effects on skeletal muscle, white adipose tissue, and liver metabolism of MICT vs. HIIT in candidates for bariatric surgery. However, the following needs to be supplemented:

1. Introduction

There are many previous studies that analyzed various factors, including obesity-related indicators, after performing MICT and HIIT in obese people. While presenting the results of these previous studies, please additionally describe the need to analyze the effects of MICT and HIIT for candidates for bariatric surgery.

2. Results

I think the results were organized well.

3. Discussion

From the discussion, it appears that the effect of MICT was significant on muscle and liver, while the influence of HIIT was significant on fat tissue. I think it is necessary to discuss the cause of why this difference appears.

4. Conclusion

In conclusion, it was suggested that while MICT had a significant effect on muscles, HIIT had a significant effect on liver and adipose tissue. I think this is different from the results. Why was this conclusion drawn?

Author Response

Dear reviewer,

Round 2

Reviewer 2 Report

Comments and Suggestions for Authors

Overall, I think the revisions and supplements have been done well. Thank you for your effort.